# Investigating the Implementation of Community-Based Stroke Telerehabilitation in England; A Realist Synthesis Study Protocol

**DOI:** 10.3390/healthcare12101027

**Published:** 2024-05-15

**Authors:** Niki Chouliara, Trudi Cameron, Scott Ballard-Ridley, Rebecca J. Fisher, Jade Kettlewell, Lisa Kidd, Leanna Luxton, Valerie Pomeroy, Rachel C. Stockley, Shirley Thomas, Adam L. Gordon

**Affiliations:** 1School of Medicine, University of Nottingham, Nottingham NG7 2TU, UK; trudi.cameron@nottingham.ac.uk (T.C.); jade.kettlewell@nottingham.ac.uk (J.K.); shirley.thomas@nottingham.ac.uk (S.T.); adam.gordon@nottingham.ac.uk (A.L.G.); 2NIHR Applied Research Collaboration (ARC) East Midlands, Nottingham NG7 2TU, UK; 3Bridges Self-Management, London SW17 0RE, UK; scott@bridgesselfmanagement.org.uk; 4Stroke Medical Directorate, NHS England, London SE1 8UG, UK; rebecca.fisher25@nhs.net; 5Department of Nursing and Community Health, Glasgow Caledonian University, Glasgow G4 0BA, UK; lisa.kidd@gcu.ac.uk; 6Northampton General Hospital NHS Trust, Northampton NN1 5BD, UK; leanna.luxton@nhs.net; 7School of Health Sciences, University of East Anglia, Norwich NR4 7TJ, UK; v.pomeroy@uea.ac.uk; 8School of Nursing and Midwifery, University of Central Lancashire, Lancashire PR1 2HE, UK; rstockley1@uclan.ac.uk

**Keywords:** stroke telerehabilitation, realist synthesis, community stroke rehabilitation services, co-production, implementation

## Abstract

Telerehabilitation (TR) shows promise as a method of remote service delivery, yet there is little guidance to inform implementation in the context of the National Health Service (NHS) in England. This paper presents the protocol for a realist synthesis study aiming to investigate how TR can be implemented to support the provision of high-quality, equitable community-based stroke rehabilitation, and under what conditions. Using a realist approach, we will synthesise information from (1) an evidence review, (2) qualitative interviews with clinicians (n ≤ 30), and patient–family carer dyads (n ≤ 60) from three purposively selected community stroke rehabilitation services in England. Working groups including rehabilitation professionals, service-users and policy-makers will co-develop actionable recommendations. Insights from the review and the interviews will be synthesised to test and refine programme theories that explain how TR works and for whom in clinical practice, and draw key messages for service implementation. This protocol highlights the need to improve our understanding of TR implementation in the context of multidisciplinary, community-based stroke service provision. We suggest the use of a realist methodology and co-production to inform evidence-based recommendations that consider the needs and priorities of clinicians and people affected by stroke.

## 1. Introduction

Stroke is one of the largest causes of adult disability worldwide [1]. Provision of specialist rehabilitation can promote recovery and independence post-stroke [2]. Access to specialist stroke rehabilitation in community settings following discharge is a recognised priority for policy and research [2,3]. Providing timely and intensive community stroke rehabilitation may be compromised by factors affecting service capacity, including low staffing levels and the resources required to travel to patients’ homes [4,5,6]. Telerehabilitation (TR) may help address some of these barriers and complement conventional rehabilitation services, including assessment, therapy, education and monitoring [7,8].

TR uses information and communication technologies ranging from telephone and internet-based video conferencing to more sophisticated digital platforms and virtual reality. Studies evaluating the effectiveness of TR for stroke survivors have shown promise. TR may augment rehabilitation intensity in the post-acute phase and help address stroke survivors’ ongoing needs [9,10,11,12]. TR has comparable effects with usual care on stroke survivors’ performance of activities of daily living (e.g., walking, dressing), and supports self-management and monitoring between face-to-face sessions [9,10,11,12]. TR may also facilitate cost-effective access to community rehabilitation for people unable to travel due to mobility issues and those living in rural, remote and under-served areas, without increasing caregivers’ burden [9,10,11,12].

The COVID-19 pandemic highlighted the importance of TR in overcoming barriers to access [8,13]. Clinical guidelines in England and other countries recommend the provision of TR as an adjunct to in-person therapy [3,14,15,16]. However, the 2021 National Stroke Audit report (SSNAP) found the adoption of TR in England to be variable and slow-paced, and emphasized the need for more progress [17]. It remains unclear how clinical teams decide whether TR is offered and to whom, and what factors influence this. TR toolkits have been developed to support clinical delivery, but we lack evidence-based recommendations to guide the implementation and integration of TR services in the context of specialist stroke rehabilitation services in England [18,19].

Prior to broad-scale dissemination, we need to understand whether TR excludes or disadvantages certain groups of people [8,20,21]. Questions remain about whether and how TR services are accessible to patients with cognitive, communication and visual difficulties, all of which occur frequently following a stroke [9,22,23,24,25]. In the trials included in the Cochrane stroke TR review, most patients were under 70 years of age, and information on participants’ digital literacy, race and other socioeconomic factors was not provided [9]. People with certain sociodemographic characteristics, such as older age, living alone and with limited access to digital equipment, may be at higher risk of digital health disparities and missing out on TR services [25,26,27]. People who experience digital exclusion in Britain tend to be older, less educated, more socially isolated and more socioeconomically disadvantaged [28]. Therefore, it is essential that efforts to inform future TR developments are guided by evidence that addresses inequalities in access [11,20,29].

In a survey of 1949 stroke survivors and family members across the UK, 17% of respondents expressed their experience of TR in negative terms, suggesting it may not be suitable for everyone [30]. Further research is needed to understand why. National and international TR studies have found that therapists may be reluctant to use the technology, and may require targeted training and guidance to develop appropriate skills [21,26,28]. The Topol review [31] identified telehealth as the technology with the highest projected impact on the NHS workforce over the next two decades, and highlighted the need to ensure clinicians understand when, how and for whom it can be used to improve care. Understanding the experiences and perspectives of staff members and stroke survivors with regard to telerehabilitation is necessary to inform the development of educational resources and recommendations for practice.

To support the implementation of complex, remotely delivered rehabilitation interventions, we need a theory-informed evidence base that describes how contextual factors and change mechanisms generate outcomes of interest [32,33]. Traditional approaches to systematic reviews cannot account for the complexity of these interactions. These questions call for a realist approach that permits the synthesis of diverse evidence types to address the how and why questions surrounding implementation [34,35].

The purpose of this paper is to present the protocol for a realist synthesis study in community-based stroke telerehabilitation. The planned study will seek to:

1.Understand how TR can be implemented to support the provision of high-quality, equitable, community-based stroke rehabilitation in England and under what conditions;2.Explore clinicians’ and patients’ experiences and perspectives of TR to identify training and support needs;3.Co-develop recommendations to inform TR implementation in clinical practice.

## 2. Methods

### 2.1. Methodological Framework

Given the variable terminology seen in the literature, we will use “telerehabilitation” as an umbrella term to refer to the use of information and communication technologies to provide the full range of rehabilitation services, including therapy (in real time or conducted independently by the patient), assessment and education [36].

The study will involve conducting a realist synthesis [34]. This type of literature review is recommended for bodies of evidence characterised by conceptual ambiguity and methodological diversity, as in the case of TR interventions [9]. To examine “how” and “why” TR achieves outcomes, this review will consider both quantitative and qualitative studies. Building iteratively on diverse sources of evidence, realist reviews (or realist syntheses) aim to articulate, test and refine programme theories that outline how the interaction between the implementation context (C), the intervention’s resources and stakeholders’ responses (Mechanisms, M) generates the outcomes of interest (O) [37,38]. Thus, realist reviews allow us to examine how the outcomes of an intervention relate to both the individuals involved and the context of its implementation.

We will assess how practitioners and patients perceive and act upon the resources offered by TR (M), and how this interaction drives the delivery of evidence-based and equitable services (O) in certain conditions and for particular groups of stroke survivors (C). The conceptual framework presented in Table 1 was informed by an initial scoping of the literature (Appendix A) and will underpin the development of our preliminary programme theories. The outcomes of interest will be informed by core components of evidence-based community rehabilitation services recommended by clinical guidelines, including the provision of intensive and person-centred rehabilitation [2]. Our exploration of context will mainly target three levels of contextual determinants: (1) service-level characteristics, including technology infrastructure and geographical location; (2) team-level characteristics, e.g., training opportunities, and (3) patient-level characteristics such as communication difficulties and digital literacy [39]. The framework will be our theoretical starting point, but it will be iteratively adapted in response to emerging literature and empirical findings.

An example of a potential Context–Mechanism–Outcome (CMO) configuration may be the following: “Within the context of an organised multidisciplinary team (Context), the provision of TR and support tailored to patients’ needs (Mechanism-Resources), will influence patients’ willingness to engage with technology (Mechanism-Responses) and determine their commitment to the rehabilitation plan (Outcome)”.

### 2.2. Study Design

The study will be undertaken across three work packages. Figure 1 presents a flowchart of our planned research activities.

#### 2.2.1. Work Package 1 (WP1): Realist Evidence Review (Months 1–12)

Rather than seeking comprehensive coverage of the evidence base, a realist review involves iterative, stepwise literature searches building on the findings from previous search. The search is systematic and the process should be documented in an explicit and transparent manner. We will follow the quality standards for conducting and reporting realist reviews [35,40]. The review will be conducted through the following interconnected stages.

Stage 1. Developing initial programme theories (months 1–4)

The review will begin with a broad background search to assess the breadth and range of available evidence. We will undertake topic-based searches of key databases (i.e., MEDLINE, PROSPERO, Cochrane library) combining the population group (stroke) and the intervention of interest (telerehabilitation) to scope the evidence and sensitise the research team to the literature [41]. We will seek to identify key works in the topic area, of any study design, that can contribute important insights towards programme theory development. Clinical guidelines and other policy documents will be particularly important to consider at this stage; it is suggested they rest on decision-makers’ assumptions about the effectiveness and implementation of a healthcare programme, and they will help us frame the “formal programme theory” regarding how stroke TR works and achieves outcomes [42].

The programme theory development process will be informed by consultation with a group of “expert stakeholders”. We will seek representation from stroke survivors and family carers, multidisciplinary professionals from community-based rehabilitation teams, researchers with relevant expertise, policy-makers and third-sector networks. Their role will be to sense-check and contextualise key messages from the literature, inform hypothesis testing, and ensure programme theories and recommendations reflect real-world experiences. Members of our Expert Stakeholder Group (ESG) will be able to meet online or face-to-face, based on their preferences. Sub-groups may be created to facilitate discussions and ensure marginalised or minority voices be heard [43]. Meetings will be audio-recorded, and transcripts will be treated as research data and serve to maintain an audit trail of the decision-making and programme theory development processes. By the end of this stage, we will have articulated and prioritised candidate programme theories, which will inform further targeted literature searches in subsequent stages.

Stage 2. Developing search strategy and searching the evidence (months 4–10)

Preliminary programme theories identified in Stage 1 will inform the search strategy and ensure it represents key concepts and phenomena of interest. A Context–Intervention–Mechanism–Outcome (CIMO structure) will be developed to describe one, two or more of the following components of the programme theory, combined using the AND Boolean operator [41]: (1) The Context or population group (e.g., stroke survivors). (2) The Intervention/programme of interest (e.g., telerehabilitation). (3) A suggested Mechanism (e.g., clinicians’ perceptions of telerehabilitation). (4) An Outcome of interest (e.g., intensive rehabilitation practice).

Electronic searches of the major health, social and welfare databases will be conducted. These will comprise but not be restricted to: EMBASE, MEDLINE, CINAHL, PsychINFO, Web of Science, and Cochrane Database of Systematic Reviews. Regular alerts on the major databases will be set to keep abreast of the rapid advancements in the area. We envisage that the initial search will include all types of empirical studies and reviews as well as grey literature including policy documents, guidelines and service specifications. We will exclude: (a) studies published before 2010, to ensure our findings reflect a contemporary landscape of healthcare provision and technology use, (b) evidence not available in the English language, and (c) literature relating to children (<18 years).

Reflecting the iterative nature of the realist approach, eligibility criteria may expand to further focus the searches in light of emerging data as the review progresses [40,41]. For instance, a subsequent search may target the exploration of a particular mechanism (e.g., therapists’ attitudes towards technology) and its influence on outcomes of interest (e.g., adoption of stroke telerehabilitation) within a particular context (community stroke rehabilitation services).

Complementary search techniques including citation tracking, snowball sampling [44], and contacting authors will be used alongside database searches. Cluster searching [45] will involve using a key work in the topic area as a retrieval point of related research outputs, which may help inform theory development. Guidance from our research team and stakeholder expert group will maximise opportunities to identify relevant evidence.

Search results tables will be developed to capture information on the source and characteristics of each study, including the database name, coverage dates, studies’ aims, methods, a description of the intervention (where relevant) and key findings, along with comments on the reasoning behind their inclusion or exclusion [46]. A study flow diagram will be developed to summarise the search and selection of studies through both database and manual searching and record any changes in the direction of searching. The realist approach does not require the search to be exhaustive but, in line with the principles of theoretical saturation, searches will be completed when additional data do not add to or contradict candidate programme theories [41].

Stage 3. Quality appraisal and data extraction (months 5–12)

Search results will be initially screened by title and abstract. For studies that meet our eligibility criteria, data quality appraisal of full texts will be undertaken. The same quality criteria will be applied to all study types and will involve assessing evidence for: (a) its relevance for theory testing (i.e., whether it addresses the programme theories under assessment), (b) the rigour with which it has been produced (i.e., whether the methods used to generate the relevant data are credible and trustworthy) and (c) its richness (i.e., the extent to which rich, explanatory descriptions of the findings are provided) [35,47]. For each study judgements will be recorded and a rating system will be devised (e.g., high/medium/low relevance) to enhance the transparency of the process and promote consistency between reviewers. Rigour will be strengthened by having 10–20% of the screened results checked by other members of the research team.

Data extraction and analysis will be informed by the process outlined by Rycroft-Malone [48]. A coding framework will be iteratively developed to reflect concepts of interest (see Table 1 for examples) and the programme theories under evaluation. Direct excerpts with relevant information will be extracted from each article and into the NVivo 14 software for organisation and analysis. For quantitative studies, rather than extracting numerical data, descriptions of key findings and authors’ interpretations will be captured. Extracted data will be organised thematically under relevant headings informed by the coding framework. Additional headings corresponding to new themes and sub-themes may be added as emerging findings clarify, refine, or contradict the predefined framework. The next step will involve looking for connections across themes to start forming CMO configurations. A record of the process will be kept so that each CMO can be linked back to the source documents and a judgement can be made about the characteristics and quality of evidence underpinning them.

Throughout the process, we will keep detailed records of our actions and decision-making processes, and maintain reflexivity. The list of included/excluded studies and the data extraction forms will be reviewed at monthly project meetings. By the end of WP1 we will have drawn key lessons from the evidence to address our first study aim. In WP2 we will draw on the feedback of clinicians, stroke survivors and family carers to further test and refine programme theories and assess their relevance in real world settings, so that the findings can inform the development of recommendations for wider use.

#### 2.2.2. Work Package 2 (WP2): Testing and Refining Programme Theories through Primary Data Collection (Months 10–17)

We will conduct realist-informed interviews [49] and focus groups [50] with service users, carers and clinicians. Participants will be recruited from three community stroke rehabilitation services in England. Site and participant selections will be guided by purposive sampling strategies, to ensure we include information-rich cases in relation to concepts identified as important in WP1 [51,52]. Based on an initial scoping of the evidence, we expect that site selection should capture variation in relation to: (a) the level to which TR was adopted (based on National Stroke Audit results), (b) the rurality of the service location [53] and (c) the level of deprivation.

##### Service User Interviews

Up to 30 interviews with patient–family caregiver dyads will be undertaken over 3 sites (N = ≤60 participants) [54]. A purposive sampling framework will be developed to ensure diversity in relevant characteristics (e.g., ethnicity, gender, age, severity of disability) and representation from patients who declined TR.

All patients with a stroke diagnosis who are (1) actively on the services’ caseloads at the time of recruitment, (2) medically stable, and (3) able to give informed consent will be eligible for inclusion. We will not exclude patients based on severity/type of disability or sociodemographic characteristics.

Staff members will identify potential participants, based on our sampling criteria, and make the first approach. Semi-structured interviews will be conducted, face-to-face or via video-calls depending upon participant preferences. They will last up to 45 min, though we will allow longer for participants who need it (e.g., in the presence of communication difficulties).

Interview schedules will be informed by WP1 findings. In consultation with our Patient and Public Involvement & Engagement (PPIE) group and our co-authors (SBR, ST), appropriate information and consent material will be developed, including aphasia-friendly information sheets. Adjustments to the interview process, such as the use of pictorial information and visual analogue scales, will be considered [55]. We will consult with the patient or their family/carer before the interview to ascertain what methods they are most comfortable with. For patients who do not speak English, relevant materials (e.g., participant information sheet, consent form) will be translated and professional interpreters will support with the informed consent process, and will attend the interviews if required.

##### Staff Members Focus Groups

Up to 6 focus groups over 3 sites will be conducted with a cross-section of professionals from each multidisciplinary team (MDT) [40]. We will aim to understand local TR delivery and invite participants to reflect on key theories identified in the review.

Interviews will be recorded, transcribed and analysed by the research team using a framework approach [56]. The programme theories and related CMO configurations identified in previous stages will act as an overarching framework to guide the analysis. However, we will also actively seek examples confirming or disconfirming our theories.

##### Data Synthesis

Each programme theory and associated CMOs will be reviewed against evidence from the literature and the interviews. Primary data will help us test the integrity and explanatory power of programme theories brought forward from WP1. They may also serve to complement insights from the literature or highlight evidence gaps that should be addressed by future research. In addition to identifying consistent patterns, synthesis will involve capturing differences and contradictions across datasets, leading to theory refinements or revisions [42].

The generated theories will be considered in relation to implementation frameworks and mid-range theories (e.g., Normalisation Process theory [57], PERCS Framework [58]) to examine their transferability potential across similar interventions and contexts. The finalised set of CMOs and programme theories, supported by relevant examples from the datasets, will be discussed with and validated by the ESG and co-investigators. The output of this stage will be a theoretically grounded framework that synthesises insights from WP1 and WP2, and best explains how TR can support the provision of evidence-based and equitable stroke rehabilitation. Key messages for clinical practice will be extracted to inform the development of actionable recommendations in WP3.

#### 2.2.3. Work Package 3 (WP3): Development of Recommendations and Dissemination Activities (Months 16–24)

The key aim of WP3 will be to “translate” findings from previous work packages into recommendations for clinical practice. We will also set the processes in place for disseminating widely and supporting an ongoing dialogue between our network of stakeholders in relation to the stroke TR agenda.

In collaboration with Bridges Self-Management Social Enterprise (https://www.bridgesselfmanagement.org.uk/ accessed on 1 May 2024), we will plan and organise two co-design events to facilitate the development of recommendations. Their structure and delivery will be informed by an experience-based co-design (EBCD) approach [59,60]. EBCD offers a method and a process to enable a diverse group of people to work together in designing quality improvement in healthcare.

We will seek representation from key stakeholders including community stroke rehabilitation teams, commissioners and Integrated Stroke Delivery Networks leads (ISDNs), third-sector organisations, stroke survivors and families. Our ESG members and research participants in WP2 will also be invited to attend. The first event will involve reflecting on study findings and prioritising key learnings with regard to TR implementation that should inform the development of recommendations. A short animation film summarising our main findings will be presented, and the study participants will be invited to share their experiences of TR, illustrating good examples of practice and areas for improvement.

The second event will explore potential strategies towards improving the staff and patient experience, and produce a first iteration of co-developed recommendations relevant to a range of stakeholders (community stroke teams, ISDNs, stroke survivors and families) and feasible for implementation in the NHS context. Appropriate formats and avenues to maximise the reach and impact of recommendations will also be discussed.

An accessible report summarising the outcomes of the group work will be circulated to the contributors. A final online event will explore the development of a community of practice—a network of stakeholders with a specialist interest in stroke TR. We envisage this community to offer an opportunity for ongoing feedback on the applicability and transferability of our findings, and represent a platform for sustained engagement and communication between researchers, clinicians, policy-makers and service-users [61].

### 2.3. Stakeholder Involvement

We will adopt a participatory approach to actively involve stroke survivors, family carers and rehabilitation professionals, throughout the research process. We believe this to be key in achieving the aims of this study and ensuring recommendations are usable and reflect service users’ needs.

People with stroke have been involved throughout the development of this protocol, providing independent advice and guidance. The study has been discussed with the Nottingham Stroke Research Partnership group, an established PPIE group of stroke survivors and carers, who were supportive of the study’s aims and helped frame the research questions and address design issues. The group will continue to be consulted at key stages of the project, including recruitment and dissemination strategies. A group member will also join the study steering group to provide advice and monitor progress. Co-author SBR, a stroke survivor and expert in co-production, contributed to the design of WP3, and will facilitate the delivery of the co-design workshops.

In WP1, people affected by stroke and rehabilitation professionals will be invited to join the expert stakeholder group (ESG) as partners in knowledge production. As research participants, in WP2, stroke survivors and clinical staff will be recruited to ensure we capture their experiences and perceptions of TR, as well as their support/training needs. Their accounts will complete knowledge gaps, contextualise insights from the literature review and inform the evidence synthesis. As part of WP3, ESG members and research participants will be invited to attend the co-design events with a view to: (1) receiving feedback on findings, (2) sharing their experience of TR, and (3) contributing to the identification of priority areas for recommendations. We will offer opportunities for them to continue being part of the academic and stroke TR community, should they wish to.

We will take active steps to reach out to and engage with groups of people previously under-represented in stroke TR research, including ethnic minority groups and people who may experience digital exclusion. We will capitalise on our links with the National Centre for Ethnic Health Research as well as local community groups to inform our recruitment strategy and facilitate the involvement of “seldom heard” groups. Accessibility issues have been considered in our budget, including funds for translation/interpretation services and reimbursement for people who do not have internet (Wi-Fi) access and require using their mobile data to participate in this study.

## 3. Discussion

### 3.1. Contribution to the Literature

This paper describes the protocol of a realist synthesis aiming to explore how telerehabilitation (TR) can support the provision of evidence-based and equitable stroke rehabilitation in the community. The recently updated NICE guidelines suggest that stroke TR is at least as effective as face-to-face care in improving performance on activities of daily living and physical function. Yet, programme effectiveness does not guarantee adoption and implementation [32], and this protocol highlights the need to shed light on issues pertaining to the translation of TR into routine care in England.

The realist conceptual framework developed as part of this protocol emphasises the role of context in triggering the processes generating the outcomes of interest. It draws attention to three domains of contextual determinants at the patient, MDT and service level, and the need to understand how they influence the implementation of evidence-based TR. To uncover the mechanisms driving implementation, capturing clinicians’ and stroke survivors’ perceptions and responses to TR will be key. Stakeholders’ acceptance and readiness for TR has been previously identified as a factor influencing implementation success [62]. The theory development process will also consider informal caregivers’ views and experiences, responding to recommendations for further research on the role of family carers in stroke TR. As English et al. [11] pointed out, if TR is to be further developed, we need to ensure it does not disadvantage people who do not have access to social support.

Previous studies have noted the challenges of systematically reviewing the TR evidence base, mainly due to the breadth and great heterogeneity of the interventions and parameters evaluated [9,21]. The realist methodological approach described in this paper suggests a shift in focus from the specific characteristics of particular interventions to the underlying mechanisms and processes through which TR can achieve core aspects of evidence-based care. For instance, rather than focusing on one form of technology or discipline-specific TR interventions, we will investigate how telerehabilitation can be implemented as part of multidisciplinary service provision.

In response to concerns that TR may exacerbate the so-called digital divide [9,20], the study will draw attention to the need for a systematic approach in understanding disparities in stroke TR. In developing the protocol, we worked in partnership with a diverse group of PPIE representatives to ensure EDI considerations informed the study aims and were embedded in the study research design. Working in partnership with the communities the services are intended to benefit is key in addressing healthcare inequalities [63]; our inclusive recruitment strategy and continued work with PPIE groups and third-sector organisations will help capture the opinions and experiences of “seldom heard” groups of stroke survivors, and involve them in shaping recommendations.

### 3.2. Strengths and Limitations

Rather than aspiring to statistical generalisability, realist synthesis seeks to achieve theoretical transferability. Developing programme theories at a higher level of abstraction is suggested to promote the applicability of findings to similar contexts and families of interventions; in this case, stroke TR services [48,64]. As part of the iterative literature searches, we may draw on findings from studies in related fields to address any gaps in the stroke-specific evidence base and assess the transferability of our finalised programmes’ theories to different healthcare contexts (e.g., Hasan et al.’s study in the US context) [65].

Another strength of our methodological approach relates to the opportunities it presents for collaborative working and co-production with a diverse group of stakeholders [65]. We envisage that high stakeholder engagement will increase the potential of our findings to be relevant and usable. The ESG will have diverse membership, and will help ensure that service-users contribute to making sense and articulating key messages from the literature. ESG involvement is suggested to enhance the rigour and transparency of the theory-development process, improving public trust in research, and strengthening its potential to inform service delivery [34,35].

Our recruitment strategy will not exclude patients based on age or severity/type of disability, and a purposive sampling approach will ensure we capture key characteristics of interest (e.g., patients who refused TR, services covering rural locations). It is suggested that the sample specificity (i.e., purposive sampling approach addressing specific aspects of variation), the inclusion of individuals under-represented in previous TR research (e.g., communication difficulties, ethnic minority groups), and the strong theoretical basis (i.e., realist methodology) will enhance the information power of our sample [66]. We acknowledge, though, that our sampling framework will not exhaust all potential causes of variation in patient experience, and that our participants may not represent the wider population. Quantitative research designs will be required to evaluate the impacts of these characteristics on telerehabilitation delivery and effectiveness for defined groups of stroke survivors, and our findings may inform this future line of work.

The realist approach to evidence review prioritises an in-depth examination of certain aspects of a phenomenon over comprehensive literature coverage. Ongoing feedback from ESG along with primary data collection will help illuminate and complement insights from the literature review. We will acknowledge and reflect on the implications of evidence gaps for our findings and the future research agenda. We will need to ensure that we complete the evidence review on schedule, while keeping up with a fast-changing landscape. Our ESG and co-applicant groups comprising research, clinical and ISDN representatives may help us identify the latest research and policy developments, and inform the review process where appropriate. The iterative nature of the realist approach will provide further opportunities to identify new evidence in a timely manner.

### 3.3. Expected Findings and Implications

Understanding clinicians’ attitudes, confidence and willingness to use telerehabilitation will be a key output of the research outlined in this protocol, and may inform the design of targeted training and educational interventions to facilitate implementation. The findings may also help identify groups of people who are disadvantaged in relation to accessing and benefiting from TR, and alert researchers and clinicians to potential barriers to equitable TR service provision.

There is great variation in how telerehabilitation is defined in the literature, and this study may contribute to achieving conceptual clarity in the context of stroke care. Developing evidence-based theories of the way TR works and achieves change will promote a shared understanding of how TR interventions are expected to generate outcomes, and inform the design of quality improvement projects and evaluative studies. Future research could capitalise on the knowledge of key contextual determinants influencing TR implementation to explain variations in effectiveness, and help to design theory-informed, targeted interventions.

The study responds to stroke research priorities and policy recommendations for the improved organisation and delivery of high-intensity community stroke rehabilitation services [3,67,68]. In England, the newly introduced National Integrated Community Stroke Service model (ICSS) proposes the use of TR to complement face to face rehabilitation, where appropriate, but also calls for the further evaluation of outcomes and patient experience [69]. The findings from this study could inform these efforts to improve access to specialist intensive rehabilitation services in the community through the use of TR.

The pandemic has provided rich evidence about the real-world implementation of TR. However, the great variation in the adoption and sustained use of TR after the pandemic underlines the need to understand how TR can best be deployed and supported. This study is expected to highlight evidence gaps and priorities for research, aligned with practitioners’ and service users’ needs. Developing a community of practice with a specialist interest in TR will provide opportunities for knowledge exchange, nationally and internationally, with the view to driving forward the stroke TR agenda.

## Figures and Tables

**Figure 1 healthcare-12-01027-f001:**
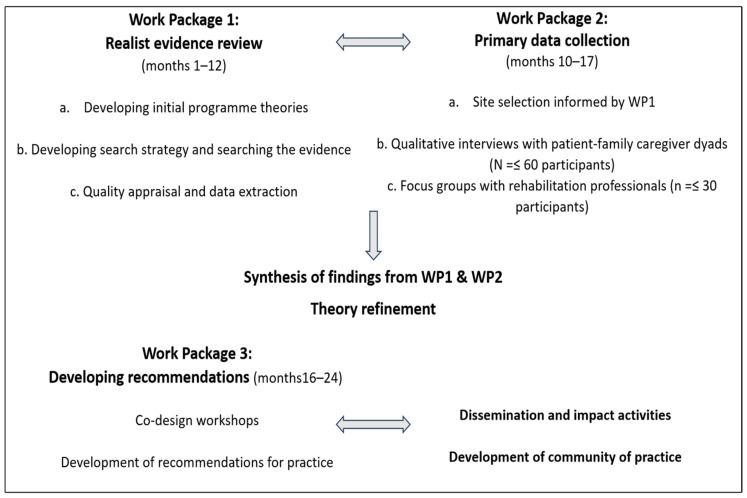
Study flowchart.

**Table 1 healthcare-12-01027-t001:** Programme theory framework.

Context	Mechanisms		Outcomes
	Resources	Responses	
Service level factors (e.g., service capacity, geographical location, technology infrastructure) Team level (e.g., multidisciplinary approach, training opportunities) Participant level (e.g., communication/cognitive difficulties, living arrangements, age, digital literacy)	Remote delivery of therapy/assessments/information provision Synchronous/asynchronous delivery Personalised, goal-oriented rehabilitation	Staff perspectives/behaviour Stroke survivors/family carer’s perspectives/behaviour	Rehabilitation intensity Person-centred care Equitable access to rehabilitation

## Data Availability

No new data were created or analyzed in this study. Data sharing is not applicable to this article.

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
