# Peer review of "Investigating the Implementation of Community-Based Stroke Telerehabilitation in England; A Realist Synthesis Study Protocol"

_healthcare, 2024, doi:10.3390/healthcare12101027_

Round 1

Reviewer 1 Report

Comments and Suggestions for Authors

Hello

In this article, the authors propose a new protocol to understand how TR works and what population can benefit from it. However, the protocol has not yet been implemented, so it remains as future work. As a result, I recommend that the authors modify the title of the manuscript because the word “understanding” suggests that the primary outcome of the study will be an explanation of the underlying mechanism governing TR, which is not the case. In the same sense, the abstract indicates that the paper will contain a results section which does not exist (I agree that a potential counter argument based on the meaning of the word "will" could elucidated; but please take this comment as a friendly invitation to improve the clarity of the manuscript).

Here a few additional recommendations:

1. Modify the title and abstract to better reflect the actual purpose of the paper, which is to present the proposed protocol.

2. Differentiate the purpose of the paper from the purpose from future studies applying the protocol.

3. The results section should present the protocol, which is the main results of the study, and the methods sections the process to arrive to the protocol.

Finally, I believe the article has sufficient merit to be considered for publication after addressing the comments above.

Author Response

We would like to thank the reviewer for taking the time to review our paper and providing their feedback. In response, we implemented a number of changes which we believe have improved the clarity of our manuscript.

Title: We rephrased the title to ensure it clearly conveys key information about the article i.e. that it presents the protocol of a realist synthesis investigating the implementation of home-based stroke telerehabilitation in England.

Abstract: We have now removed the abstract headings in line with journal requirements.

Abstract & Introduction: We have revised the aims to distinguish between the purpose of this paper and the aims of the ongoing study.

Methods section: We used this section to present the methodological approach and research design of the planned study in line with the structure adopted by previously published protocols in Healthcare.

Abstract & Discussion: We have revised these sections to distinguish between direct implications from this protocol paper and implications from expected results.

Reviewer 2 Report

Comments and Suggestions for Authors

METHODS

1. In relation to the inclusion of qualitative and quantitative studies, as they usually present results that are very different from each other, it would be of interest that the authors explain as exhaustively as possible how the results are going to be managed and interpreted.

2. It is of interest to establish the PICOS strategy, key words and MESH terms used.

3. Similar protocols from other populations or with similar  methodologies should be taken into account to discuss with them.

4. How is this sample number of patients to be included in the  study obtained? Is it enough to obtain significant results? What is the sample number used in previous studies? Or have you done a statistical analysis? In this case, the process should be clarified and detailed.

5. In relation to age, it is given importance during the introduction  and development of the article, however it does not seem to be taken into account in the inclusion and/or exclusion criteria, during the methodology or during the discussion.

DISCUSSION

1. In the discussion there are hardly any references, this should change and compare more with similar studies or studies that intend to do similar things in another population.

2. It would be of interest to remember the objective of the study prior to discussing with other articles.

3. More than a discussion, it is a conclusion of expected results and what this study would contribute. I recommend rewriting this section being more critical and taking into account the current evidence.

REFERENCES

1. They are not in the proper format

Author Response

We would like to thank the reviewer for taking the time to review our paper and providing their feedback. In response, we implemented a number of changes which we believe have improved the clarity of our manuscript.

METHODS

R2.1. In relation to the inclusion of qualitative and quantitative studies, as they usually present results that are very different from each other, it would be of interest that the authors explain as exhaustively as possible how the results are going to be managed and interpreted.

Response: Both qualitative and quantitative studies will be appraised for their ability to contribute to theory development and relevant information, in the form of a direct excerpts from the articles, will be extracted and organised thematically. For quantitative studies, rather than extracting numerical data, descriptions of key findings and authors’ interpretations will be captured. We have expanded on the relevant section in our Methods to clarify the process of data extraction and analysis.

 R2.2. It is of interest to establish the PICOS strategy, key words and MESH terms used

Response: In line with the realist methodology we envisage that a CIMO (rather than a PICO) strategy may be useful in informing the development of search queries and have added the following text:

Preliminary programme theories identified in Stage 1 will inform the search strategy and ensure it represents key concepts and phenomena of interest. A CIMO structure will be developed to describe one, two or more of the following components of the porogramme theory combined using the AND Boolean operator (Booth, 2018): 1. The context or population group (e.g. stroke survivors), 2.  the intervention/programme of interest (e.g. telerehabilitation), 3. a suggested mechanism (e.g. clinicians’ perceptions of telerehabilitation) and 4. an outcome/ phenomenon of interest (e.g. intensive rehabilitation practice).

To better illustrate examples of keywords and search terms we have now added, as a supplementary file, the search terms used as part of the initial scoping review that informed the conceptual framework outlined in Table 1.

R2. 3. Similar protocols from other populations or with similar methodologies should be taken into account to discuss with them.

Response: Our aim is to inform the study and implementation of stroke specific telerehabilitation services in the community. We have only identified one review protocol focusing on stroke telerehabilitation, which is now referenced in the discussion. A comprehensive account and critical appraisal of study protocols in other populations is beyond the scope of this study.

R2. 4. How is this sample number of patients to be included in the study obtained? Is it enough to obtain significant results? What is the sample number used in previous studies? Or have you done a statistical analysis? In this case, the process should be clarified and detailed.

Response: There is widespread agreement among qualitative methodologists that there is no set number of interviews that can be assumed to achieve data completeness and saturation. Realist synthesis aspires to theoretical transferability rather than statistical generalisability and we have expanded on the relevant paragraph in the Discussion section. As such, the emphasis shifts from the number people we talk to, towards an in-depth understanding of “how” our respondents have experienced telerehabilitation and how these experiences compare to our hypotheses about the implementation of stroke telerehabilitation.

The following has been added in the Discussion to highlight the focus in obtaining information-rich data from purposively selected cases:

"It is suggested that the sample specificity (i.e. purposive sampling approach addressing specific aspects of variation), the inclusion of individuals under-represented in previous TR research (e.g. communication difficulties, ethnic minority groups), and the strong theoretical basis (i.e. realist methodology) will enhance the information power of our sample [Malterud,2016].

 R2. 5. In relation to age, it is given importance during the introduction and development of the article, however it does not seem to be taken into account in the inclusion and/or exclusion criteria, during the methodology or during the discussion.

The potential influence of patient-level characteristics (e.g. age, presence of cognitive, language, sensory impairment, digital literacy) on the implementation of telerehabilitation has not been clarified in the literature. In the Introduction we refer to previous reviews, which have highlighted the problem and suggested that older people and stroke survivors with cognitive, communication and visual difficulties may be under-represented in stroke telerehabilitation research.

As outlined in our conceptual framework (Table 1), we will consider how these characteristics may influence implementation, drawing on insights from previous research and primary data collection (i.e. interviews with patients/clinicians). With regards to the data collection, a realist sampling framework was developed to purposively capture characteristics of interest and ensure they are represented in our sample. As clarified in our Methods section, these characteristics include age, ethnicity, gender, digital literacy and representation from patients who declined TR (please see highlighted  paragraphs in Methods). We will not exclude patients based on severity/type of disability or other sociodemographic characteristics including age.

In our Discussion, we acknowledged that quantitative research designs will be required to fully evaluate the impact of these characteristics on telerehabilitation delivery and effectiveness but our findings may inform this future line of work.

 DISCUSSION

R2.6. In the discussion there are hardly any references, this should change and compare more with similar studies or studies that intend to do similar things in another population.

R2.7. It would be of interest to remember the objective of the study prior to discussing with other articles.

R2.8. More than a discussion, it is a conclusion of expected results and what this study would contribute. I recommend rewriting this section being more critical and taking into account the current evidence.

In response to the above recommendations we have rewritten/ re-organised our discussion section. We have also added subheadings to clarify what each sub-section refers to and distinguish between direct implications from this protocol and expected outcomes from the planned study. Our discussion section now includes: 1. the contribution of this protocol study to the literature, 2. strengths and limitations of our design and methodological approach, 3. expected findings from the study and their implications for research and practice. We have expanded to discuss each point in more detail and in the light of available evidence while making sure we do not make unsubstantiated claims, considering we are presenting the protocol for a study which has not yet generated data.

Round 2

Reviewer 2 Report

Comments and Suggestions for Authors

The comments have been answered appropriately and I believe the corrections are correct